# Body Scan Processing, Generative Design, and Multiobjective Evaluation of Sports Bras

**Audrey Bosquet [1,\*,†], Caitlin Mueller [2] and A.E. Hosoi [1,]**

[1]   Mechanical Engineering, Massachusetts Institute of Technology, Cambridge, MA 02139, USA; peko@mit.edu
[2]   Architecture: Building Technology, Massachusetts Institute of Technology, Cambridge, MA 02139, USA; caitlinm@mit.edu
\*   Correspondence: abosquet@mit.edu
†   Current address: 77 Massachusetts Ave, 3-237 Cambridge, MA 02139, USA.

**Abstract:** Sports bras are critical to the comfort and performance of female athletes, yet mechanical models of sports bras are generally not used to guide their design. Typically, assessing any sports bra's performance requires time-consuming and expensive biomechanical testing, which limits the number of designs considered. To more broadly advance knowledge on how different design properties of sports bras affect their performance, this paper presents a new design framework to explore and evaluate the sports bra design space. The framework incorporates methods for body scan analysis, fast simulation, design generation, and performance evaluation. Using these methods together enables the rapid exploration of hundreds, or thousands, of designs—each one having been evaluated on key metrics related to sports bra performance, namely, range of motion and average pressure. With this framework, designers can potentially discover a diverse set of new, high-performing sports bra concepts, as well as gain insights into how design decisions affect performance.

**Keywords:** sports bras; design space exploration; generative design; breast simulation

## 1. Introduction

### 1.1. Sports Bras and the Impact of Breasts

The sports bra is an essential piece of equipment for women, and yet it is a product that was first invented only 45 years ago. In 1975, the first sports bra was made by sewing two jockstraps together. By comparison, the jockstrap was invented a hundred years earlier in 1874 [1]. Sports bras have come a long way since 1975 and the industry continues to grow [2], but there is still room for innovation and improvement, as women continue to experience mastalgia (breast pain) during exercise [3].

Beyond discomfort, breasts can be a significant barrier to sport for women, especially for those with greater breast volume [4]. Women who become athletes regardless of this barrier may still find their breasts negatively affect their performance—a 2012 study of the London Marathon showed that women with larger breasts on average finished slower than their smaller-chested counterparts [5].

### 1.2. Sports Bra Design Process

In a typical design process, in the early stages, many different design concepts are explored in parallel [6]. As the design process progresses, the number of design concepts is narrowed down at the same time as the specificity of those designs that remain increases. In the later stages of the design process, a singular concept has been chosen, most of the major decisions have already been made, and all that is left to decide are details. It is typically in this later stage of the sports bra design process—after many stylistic decisions have been made—that performance is measured and optimized.

Design of sports bras relies heavily on the expertise and experience of the designer, and often involves multiple rounds of "trial and error" [7]. A series of try-on tests with live models informs how paneling can be adjusted in order to achieve a better fit. If designers wish to improve the support of a sports bra, they must conduct biomechanical experiments (motion capture studies on subjects) to compare breast movement in different bra prototypes. Biomechanical testing for bra evaluation is time-intensive and costly, which can limit the number of bra designs that can be tested. As breast movement and bra mechanics are both complex systems, it is difficult to map insights from motion capture studies to design properties of the bra.

Because the evaluation of sports bras is challenging and conducted at later stages of the design process, optimization of characteristics such as support and comfort is usually limited to small variances within a selected design. Research on comparison of sports bra designs for performance goals has been either very focused, evaluating already developed design features against each other [8] (i.e., is a cross-back better than regular straps), or very broad, showing that "high support" bras reduce bra–breast forces compared to a "low support" counterpart [9]. The infinite field of potential designs of sports bras remains a mostly untested design space as so few designs get evaluated. As a result, there is little quantitative knowledge on how to design sports bras for desired outcomes.

### 1.3. Evaluating Designs: Performance

For other sports equipment, such as shoes, the desired outcome, or "performance", of the equipment might be measured by the increased athletic performance of the wearer (how much faster they run, for example). Although having breasts can impact athletic performance [4,5], not enough is known about how that might be exacerbated or improved with the intervention of different sports bras. Sports bra research is not yet at a stage that allows us to prioritize *athletic* performance; instead, *perceived* performance must first be addressed.

One aspect of perceived performance can be categorized by discomfort or mastalgia (breast pain) during exercise, which was reported by over 30% of surveyed female participants in the 2012 London Marathon [3]. In another study, Haake et al. showed that strain was a reasonable predictor of perceived breast discomfort during treadmill running [10]. A simpler interpretation of this finding would be to say that increased motion of the breasts will be more likely to lead to discomfort. The same study [10] showed that breast motion during running was greatest in a bare-breasted condition (wearing no bra) and that running while wearing sports bras led to less reported breast motion and discomfort than while wearing everyday bras. This supports the position of this paper that not only is a bra an important intervention to combat discomfort, but the design of that bra impacts its effectiveness.

In a large survey performed by Bowles, Steel, and Munro in 2011, women reported overall tightness of a sports bra as a deterrent for its use [11]. This is in agreement with the authors' anecdotal observations during interviews with bra study subjects who reported disliking tighter bras. Based on these precedents, the performance goals of this research address two aspects that affect perception of sports bras: breast movement and pressure.

### 1.4. Design Space Exploration

In engineering design research in general, the field of design space exploration has proven promising for enhancing the earliest stages of the design process. In architectural and building design, for example, methods have been developed and applied to increase the diversity and performance of design concepts, using a range of computational techniques [12–14]. The basic principles of design space exploration involve design parameterization; fast simulation or analysis; and techniques for sampling, optimization, and general navigation through a space of design alternatives. While this approach has not yet been applied to sports bra design, it has the potential to amplify the diversity and quality of design outcomes, leading to innovative new options for female athletes. Design space exploration is complemented by more detailed analyses and testing methods that can be carried out on selected design concepts later in the process.

*1.5. Design Exploration and Evaluation Framework*

The research presented in this manuscript aims to increase knowledge about sports bra design and performance by providing a predictive early-stage design exploration framework. This framework (shown in the context of the existing design process in Figure 1) includes methods that the authors have developed for body scan analysis, fast simulation, design generation, and performance evaluation. With these tools, hundreds or thousands of parametrically generated designs can be virtually tested. By analyzing the performance results of parametrically designed bras, designers and engineers can discover potentially surprising, new, high-performing sports bra concepts. Insights about the effect of design decisions on performance can also be gained by examining the relationship between parameters and results.

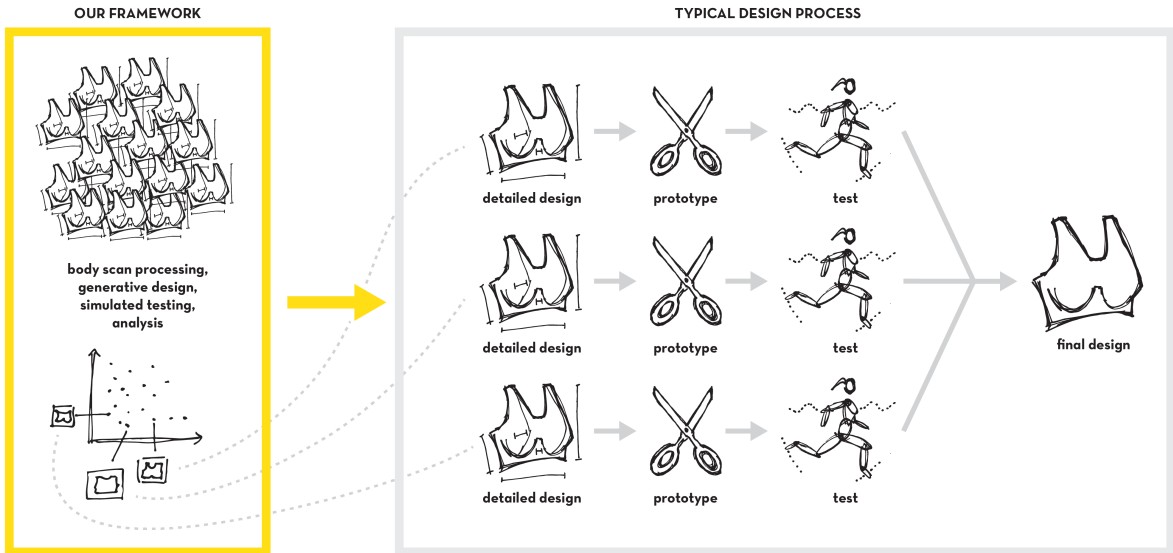

**Figure 1.** Sports bra design and proposed framework.

The use of this framework is proposed for early-stage design because although it can provide an incredible amount of information, it is not a substitute for biomechanical testing. After exploring the design space and making discoveries with this approach, designers can enter the detailed design stage more informed and confident. This increased preparation could then make the remaining design process more efficient and effective.

## 2. Materials and Methods

The proposed design exploration and evaluation framework is diagrammed in Figure 2 and described in more detail in this section. It starts with a method for body scan processing, which identifies landmarks on the scan and extracts measurements using those landmarks. Those scanned landmarks and measurements, along with six design parameters, are used as inputs to generate any number of different bra designs using a method for bra design generation. A method for breast simulation uses a force density method mathematical model and scan measurements to simulate the breast and the loads it experiences. The same mathematical model is used to test each generated bra design in a bra loading method, taking the loads observed during breast simulation as input. Finally, the outcome of those tests are evaluated. In this evaluation, the design variables of each bra are compared to their performance outcomes (in this case: range of motion and pressure) in order to identify relationships between them and find high-performing designs.

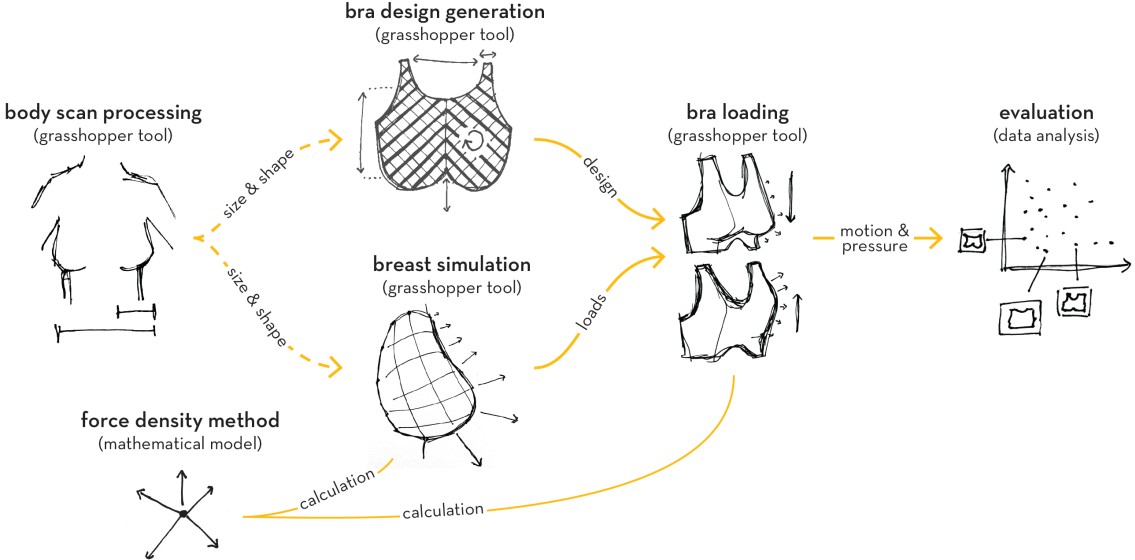

**Figure 2.** Diagram of methodology.

### 2.1. Body Scan Processing

As the goal of this research is to create a new method for designing and evaluating sports bras, a prerequisite for that is understanding the design parameters and variables necessary for creating a well-fitting garment. While detailed guidelines exist for creating multitudes of bra designs [15], it is hard to know where those guidelines come from or how well they represent the intended wearer. These guidelines are also very prescriptive, each depending on the type of bra being made.

Multiple precedents have shown that the proper fit of a bra can affect comfort, health, and athletic performance [16–18]. An overly loose bra may allow too much breast movement and acceleration during exercise, while one that is too tight can cause bruising, soreness, and can impede proper respiration. The current method of bra sizing (by measuring below the breasts, and at their widest point) involves rounding up to the nearest even number, which can lead to inaccurate sizing. There exist a few precedents for using 3D scans to inform bra sizing, including a study in 2007 which outlined a new Chinese bra sizing system based on 3D scans [19]; however, no tool is commercially available. In order to better understand what measurements would be needed for design generation, it was worthwhile to create a custom 3D scan processing method, implemented in a simple tool to carry out the research in this paper.

In addition to accepting any individual scan as data, a measurement tool should also be able to consider large sets of scans so that it can aid in the understanding of variance between shapes and sizes, and in the creation of improved sizing strategies based on real and inclusive data. As a test of its suitability for creating sizing strategies, a brief exercise of using machine learning to cluster and reorganize a large data set of body scans was performed, which will be covered in the results Section 3.1.

The intention of this research was to use results from machine learning clustering to define representative measurements to be used directly as inputs into the breast and bra simulation methods. However, this has not yet been accomplished. Instead, the breast and bra simulation (discussed in Sections 2.3 and 2.4) both take as input measurements from a single scan subject.

Body Scan Processing Method Details

The body scan processing tool was created with Grasshopper 3D, implementing contour and curvature analysis to identify landmarks and determine measurements from them. (Coltman, Steele, and McGhee showed that for large-breasted women, volumes measured from scans were most accurate when subjects lied prone [20]. Without access to prone scans, this tool focuses on consistency of

measurement over resolution.) The key landmarks are displayed in Rhino's 3D viewport and saved as an image to allow for visual verification of accuracy. Figure 3 labels these landmarks in a sample saved image from the tool.

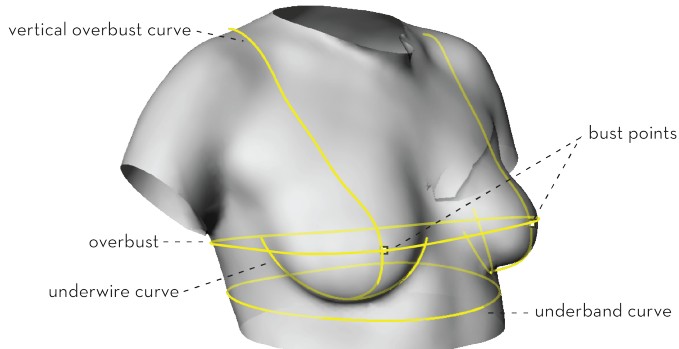

**Figure 3.** Diagram showing landmarks visualized in a sample output image.

The body scan processing tool was used to process a data set (provided by Adidas) of 583 anonymized 3D scans of women aged 18–25. (The data set of scans used for this research was provided to us (the investigators) after being recorded by a separate entity. As no identifiable information was recorded nor shared to the investigators, the use of the scans in this research was consistent with the requirements for Institutional Review Board exemption.) Aside from this overall age range of subjects, no other identifying information (weight, ethnicity) was provided. However, as shown later in Figure 15, they include a diversity of body types. The .obj mesh scans of subjects standing in an A pose (with arms held slightly away from body) are of medium quality, having some holes which were later filled by the body scan processing tool. The body scans are .obj mesh files oriented consistently, and are preprocessed to separate legs and arms from the midsection (i.e., each file had 5 meshes-2 arms, 2 legs, and 1 midsection). Of these 583 scans, the body scan processing tool was able to identify landmarks and collect measurements from 472 scans (87%), outputting 17 measurements, as well as an identifying scan number (these measurements are described in Table 1).

**Table 1.** Body scan extracted measurements.

| # | Output | Description |
|---|--------|-------------|
| 1 | Scan Number | which scan this data belongs to |
| 2 | Bust Point Distance | the distance between both bust points, where bust points are the point on each breast that protrudes the most |
| 3 | Front to Back Ratio of Overbust | ratio of the overbust circumference which encompasses breast tissue |
| 4 | Overbust Circumference | length of the overbust: the convex closed curve around the torso at the level of the bust points |
| 5 | Front to Back Ratio of Underbust | ratio of the underbust circumference which encompasses breast tissue |
| 6 | Underbust Circumference | measurement around the body, 2 cm below the underwire |
| 7 | Gore Width | distance between breasts, at endpoints of underwire |
| 8 | Breast Volume | of both breasts, in cubic centimeters |
| 9 | Underwire Height | projected vertical dimension of underwire |
| 10 | Underwire Width | dimension of underwire along the underbust curve |
| 11 | Projection Distance | shortest distance from bust point to line created by underwire endpoints |
| 12 | Bust Point to Shoulder Distance | length of vertical curve between the bust point and shoulder |
| 13 | Underbust Vertical Dimension | length of the vertical curve between the bust point and the halfway point of the underwire |
| 14 | Projection Angle | angle of the vector normal to the line created by underwire endpoints |
| 15 | Bra Height | overall vertical dimension of the bra'd area of the torso |
| 16 | Underbust Ellipse Ratio | ratio of depth to width of the best fit ellipse on the underbust |
| 17 | Band Size | traditionally (imperial method) calculated band size |
| 18 | Cup Size | traditionally (imperial method) calculated cup size |

## *2.2. Force Density Method (FDM) for Breast Simulation: Concepts*

In order to evaluate a large breadth of design parameters, simulations used to test those parameters need to be computationally inexpensive. Precedents for simulating breasts include a spring mass damper model [21] and a slightly more complex piece-wise spring mass damper [22] (shown in Figure 4), as well as a much more complex finite element method (FEM) approach [23].

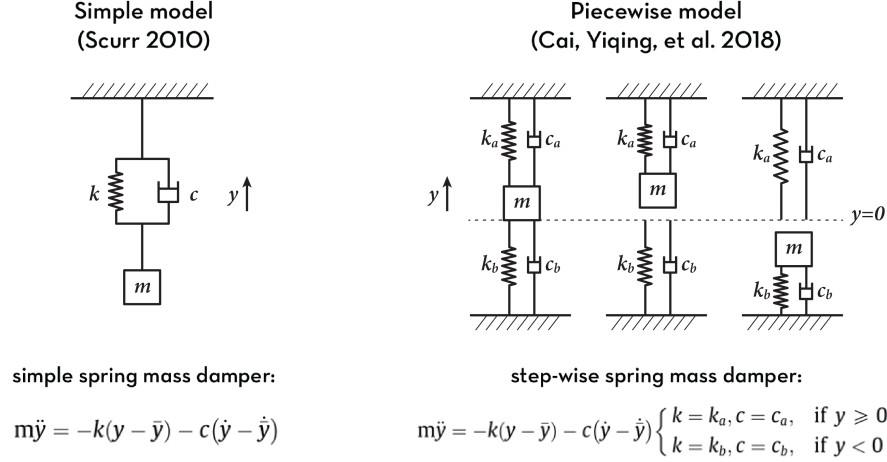

**Figure 4.** Diagrams of two precedent spring mass damper breast models.

While FEM approaches are too slow, spring–mass–damper systems, like Scurr's [21] and Cai's [22], can be incredibly fast. However, both of these models represent the breast and bra abstractly, as one point connected to one or two spring–damper systems. Although they clearly contribute to the understanding of breast motion, more information or complexity is required in order to connect the abstract system's properties to specific design parameters of a bra, such as strap width or fabric direction.

### 2.2.1. FDM Overview

Instead, the methods in this framework represent the breast-bra system using the Force Density Method. The Force Density Method (FDM), as described by Klaus Linkwitz [24], uses simple matrix operations to solve for the equilibrium of a system of bars (springs) connected by nodes under specified loads at each node. The force density (relative stiffness) assigned to each bar determines how much it will stretch. (It is important to remember that while it includes the word "force", force density in this context is not related to loads, but instead a measure of relative stiffness in each bar.) Solving the system of equations that governs the force density method problem will output a deformed surface that represents equilibrium for the given mesh under the given loads. Traditionally, this method has been used as a form-finding tool for architectural structures, particularly tensioned roofs and concrete shells. In this research, FDM is used to simulate both breast tissue and fabric in a bra. In the proposed breast simulation and bra loading methods, the FDM systems are solved using a python script that is remotely accessed from Grasshopper, which has the inputs and outputs shown in Table 2. (The python script, which is adapted from an example created by Professor Caitlin Mueller, Yijiang Huang, and Pierre Cuvilliers is included in supplemental materials.)

**Table 2.** Force Density Method (FDM) inputs and outputs.

| Inputs | Type | Outputs | Type |
|---|---|---|---|
| Base Grid Points | x,y,z Coordinates | Equilibrium Points | x,y,z Coordinates |
| Edge Node IDs | Integers | | |
| Loads at each node | Vectors | | |
| Force Densities | Number | | |
| Fixed Node IDs | Integer | | |

To fit the FDM model, the breast is considered to be a membrane, or mesh, for which each point on the surface is experiencing loads of hydrostatic pressure and each edge member of the mesh has a particular stiffness (Figure 5). In the case of the unsupported, bare breast, all force densities (relative edge stiffnesses) are considered to be equal. When the breast is in a bra, the stiffness of each edge varies depending on the bra's design.

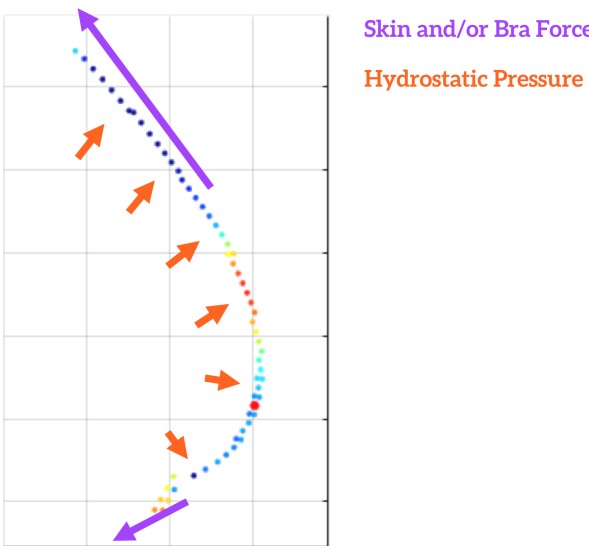

Skin and/or Bra Force

Hydrostatic Pressure

**Figure 5.** Diagram of forces considered in FDM model.

While the FDM approach is unconventional for modeling the breast–bra system, it offers the advantages of fast computational speed and easy parameterization via force densities, which control the stiffness of individual bars in the mesh network. The method has been validated on a range of other flexible mechanical systems. The authors do not suggest that it entirely replace more detailed simulation or the currently practiced physical testing methods, but that it be used in early-stage design exploration to identify designs to be studied more rigorously in later design and engineering stages.

2.2.2. Static Loading Scenarios Approximate Dynamic Movement

Spring–mass–damper models predict motion through differential equations, producing a prediction for the entire path of the breast while in motion. While a similar analysis may be possible with FDM, this research makes the assumption that instead of examining the complete path of a breast, similar knowledge is evidenced by comparing its approximate state at either end of its harmonic motion (at the extremes), producing a predicted "range" of motion. This approach further simplifies computation, allowing for greater speed (and therefore testing more designs).

Throughout this paper, the motion that is approximated in the simulated tests is that of running. In this case, the motion experienced by a breast is greatest vertically [21], so two extremes are proposed: (1) vertically upright and (2) upside-down. While these loading scenarios are likely not representative of the real motion or forces a breast experiences during running, this method prioritizes the consistency of testing designs near the extremes of their trajectory over the accuracy

of the simulation. Another limitation of this static loading method is that time-based effects (such as dampening and acceleration) are not considered. Despite these limitations, it is anticipated that static observations of approximate harmonic extremes will be suitable indicators for comparing predicted support and comfort between bra designs, especially as an informative tool for early stage design decision making.

### 2.2.3. Incompressibility

Key to accurate simulation of breasts and bras (seen in upcoming sections) is to consider the breast as incompressible which constrains its volume. Without constraining the volume of the breast, initial simulations were often undersized (demonstrated in Figure 6).

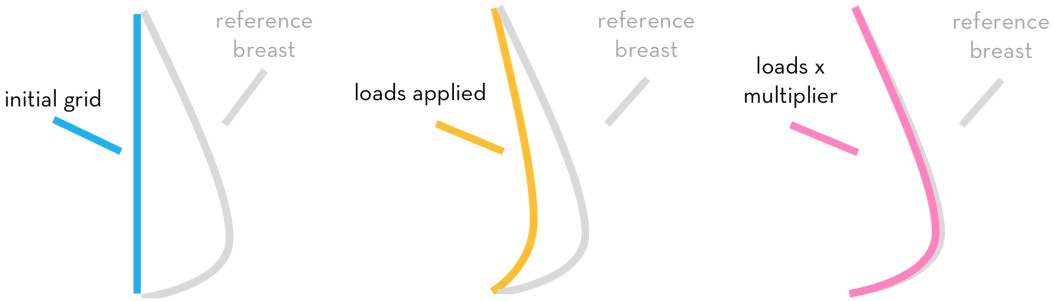

**Figure 6.** Side view diagram of applying a multiplier ($\lambda$) to FDM breast.

For more accurate simulation of breasts, the incompressible condition is met by introducing a multiplier ($\lambda_{breast}$) across all hydrostatic loads. This is analogous to using a Lagrange multiplier to enforce incompressibility of a fluid [25]. The resulting equation for pressure for both simulated breasts and bras is

$$P_{breast} = \lambda_{breast} P_{hydrostatic}$$
$$\text{such that:}$$
$$\lambda_{breast} = \arg\min_{\lambda_{breast}}(|V_{real} - V_{sim}(\lambda_{breast})|)$$

(1)

In order to calculate the correct multiplier, it is necessary to iterate through different values of $\lambda_{breast}$ until the target volume ($V_{real}$) is reached.

### 2.3. Force Density Method (FDM) for Breast Simulation: Implementation

As part of the framework, breasts are simulated in order to create realistic loads with which to test generated bra designs. As described in the Section 2.2.1, the Force Density Method is used to model the breast (and later the bra). Table 3 shows the input and output variables for the breast simulation method. As this is a novel technique, a sample bare-breasted scan is used throughout the creation of the breast simulation tool for validation. The scan is of a ~34DD sized woman.

**Table 3.** Breast simulation inputs and outputs.

| Inputs | Type | Outputs | Type |
| --- | --- | --- | --- |
| Bust Root | Geometry (curve) | Deformed Breast Shape | Geometry (points & lines)) |
| Breast Volume | Number (mL) | Loads at Each Node | Vectors |
| Rotation Angles | Numbers (degrees) | | |

The initial grid for FDM starts with a curve representing the breast boundary. The curve is patched using a Rhino/Grasshopper function called "patch" that creates a surface from one or several bounding curves. The surface is generated by finding the best fit plane for the input curve(s) and deforming it until it meets those curves. This patch defines the back wall of the breast, and is subdivided in

to a grid, creating the unloaded and undeformed mesh that will be the input for the force density method (Figure 7).

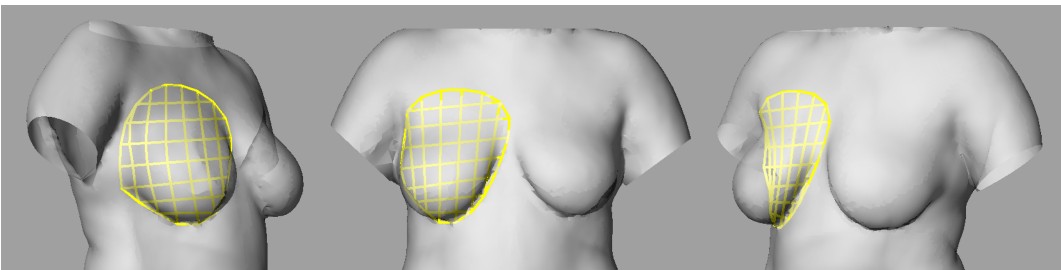

**Figure 7.** "Flat" initial breast grid for FDM.

In order to accurately simulate the breast, the loads at each vertex in the breast grid (which is representative of the skin membrane) are determined as hydrostatic and incompressible (as described in Sections 2.2.1 and 2.2.3) following the equation

$$P(i) = \lambda_{breast} \, d(i) \, \rho \, g$$
$$\text{where } d(i) = \text{depth of vertex i}$$

(2)

with the direction of load at each node equal to the normal vector of the deformed (loaded) mesh at that node. Because the normal vectors of a deformed mesh can only be determined after it has been loaded, the process needs to be iterated (or optimized) in order to approach more accurate normal vectors to input into the calculation. The breast simulation result following this process can be seen in Figure 8.

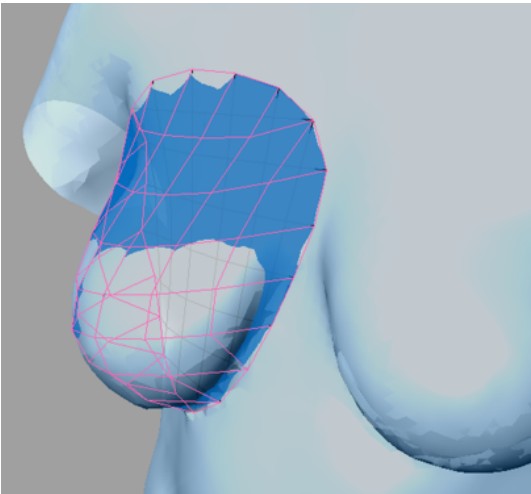

**Figure 8.** FDM-simulated unsupported breast compared to example scan.

Rotation

Now that the simulation is able to create a convincing breast shape, as seen in Figure 8 where it is overlaid on the sample bare-chested scan, the breast simulation tool can be used to (virtually) "test" bra designs. As described in the earlier section on FDM, bras will be tested by loading them with breast loads simulated at two angles with respect to gravity (examples in Figure 9). Angles are chosen to represent expected motion during the intended exercise for the designed bra.

**Vertically Upright**              **Vertically Upside-Down**

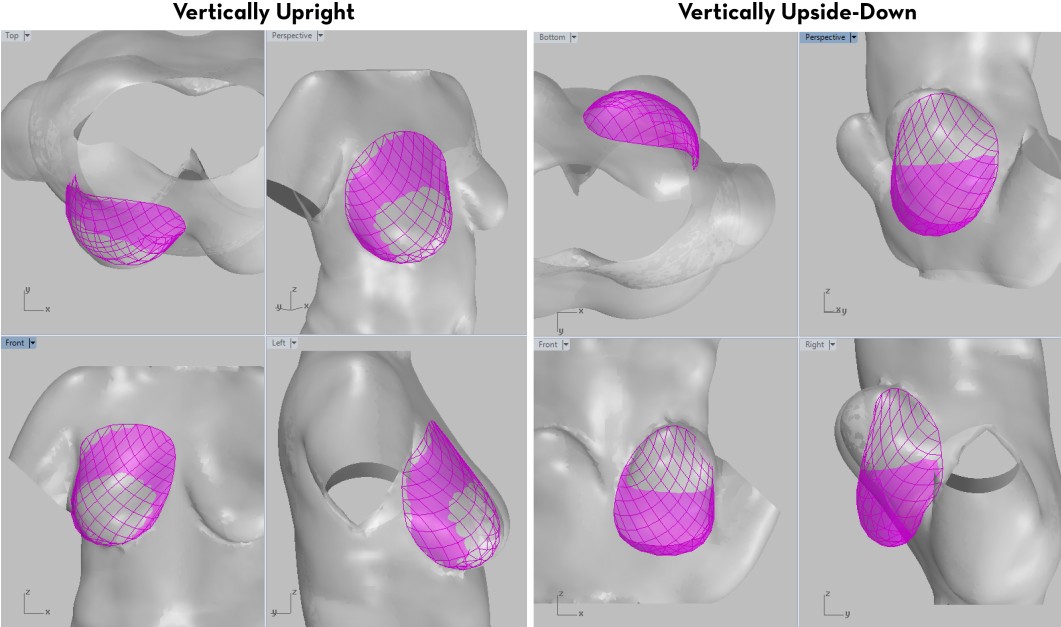

**Figure 9.** Renderings showing breast simulated at vertical extremes.

### 2.4. Breast–Bra System

#### 2.4.1. Loading Simulation

The method for relating the unsupported breast loads from breast simulation to loads impacting a bra is very direct, and therefore very simple. This method takes the loads observed at each point in the unsupported breasts and maps them directly to points in a generated bra. Although this approach may not be exact, it is a first-order approximation if slippage between the bra and the breast is minimal. Furthermore, it is likely to be appropriate for comparing performance between bras because it is applied consistently. Fundamentally, this simulation tests each generated bra's response to the same set of loads, which are an approximate reflection of breast mechanics.

Points and corresponding loads from the simulated breasts are mapped to points on the bra using nearest neighbor analysis after both geometries are projected to the same plane. Bra points that are not close enough to breast points (within a tolerance) have expected loads of zero. This can be observed in Figure 10, which shows the vectors of two loading scenarios that have been mapped to a bra design. Many of the gray bra points in the bottom center of the bra, as well as towards the top of the strap, do not have vectors mapped to them. This is because there is no expected breast tissue behind those vertices in the bra. Finally, as breast volume is still considered incompressible within a bra, the bra loading simulation also implements a multiplier ($\lambda_{bra}$):

$$P_{bra} = \lambda_{bra}\hat{P}_{breast}$$
$$\text{where } \hat{P}_{breast} \text{ are mapped breast loads}$$
$$\text{such that:}$$
$$\lambda_{bra} = \arg\min_{\lambda_{b}ra}(|V_{real} - V_{sim}(\lambda_{bra})|) \tag{3}$$

As described in the section on Force Density Method modeling and observed in Figure 10, each bra design is loaded in two separate scenarios, in order to make predictions about range of movement allowed by the bra. The scenarios can be any two angles of rotation of the breast, with respect to gravity. During the research presented in this paper, tests of the methods are conducted with the two scenarios of a breast when standing upright, and the same breast when rotated 180, such that it is upside-down.

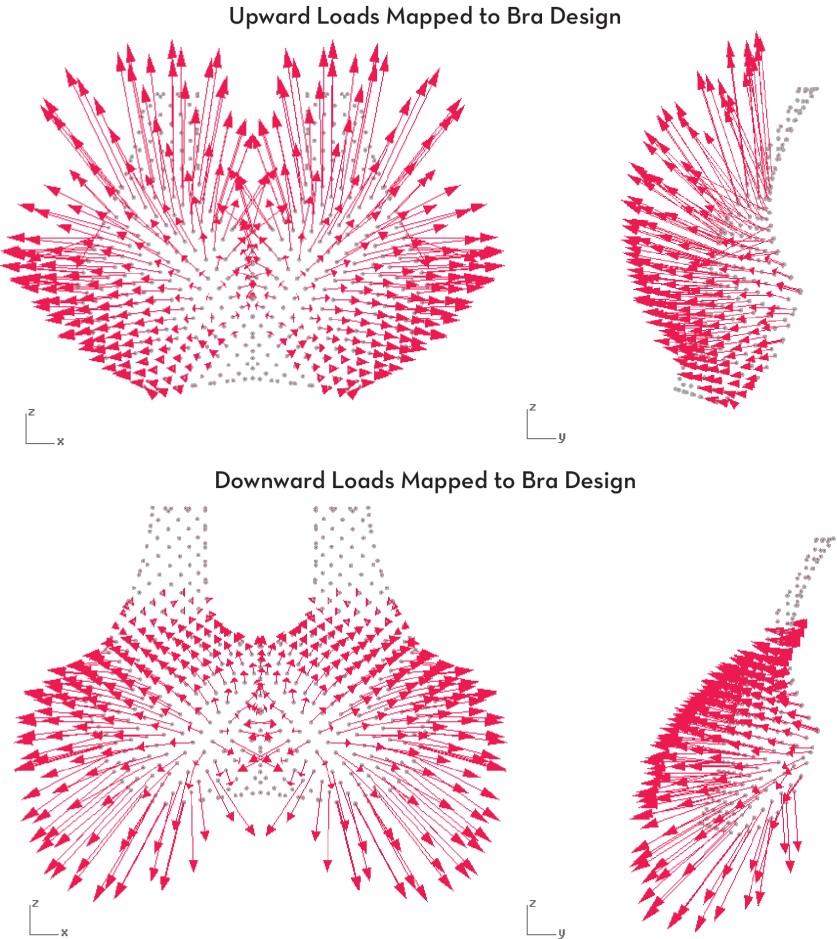

**Figure 10.** Visualization of vector loads mapped to bra from breasts.

### 2.4.2. Bra Performance Evaluation Metrics

Once a bra is tested with the load cases from the two extreme scenarios, the bra's performance is recorded for two metrics: average motion and pressure. These were chosen because they are representative of real performance evaluation metrics for bras. Reducing movement during exercise reduces breast strain, which has been shown to be a predictor for discomfort [10]. Pressure is often seen as the compromise for support, where stiffer and tighter sports bras support breasts better and reduce more movement, but they are disliked for the increased pressure that is felt in these bras [11].

Average motion (M in Equation (4)) is calculated by comparing the positions of all vertices in the bra between each rotation-specific loading scenario, summing the euclidean distance (or displacement), and dividing that sum by the number of non-fixed points.

$$M = \frac{\sum_{i=0}^{n}((x_{1_i}-x_{2_i})^2+(y_{1_i}-y_{2_i})^2+(z_{1_i}-z_{2_i})^2)^{1/2}}{n}$$
where $x_{1_i}, y_{1_i}, z_{1_i} = x, y, z$ positions of $i^{th}$ vertex for load scenario 1,
$$x_{2_i}, y_{2_i}, z_{2_i} = x, y, z \text{ positions of } i^{th}\text{vertex for load scenario 2,}$$
$$n = \text{total number of vertices}$$

(4)

Because $\lambda_{bra}$ (described earlier in Equation (3)) represents the additional effort needed to meet incompressibility of the breast, it is equated to the real and perceived pressure of the calculated condition. For example, a bra designed with stiffer members will require a larger $\lambda_{bra}$ in order for the stretched mesh to reach the target volume, compared to a looser, less stiff bra. In this example, the larger $\lambda_{bra}$ of the stiffer bra would indicate a higher pressure exerted onto the body by the bra, suggesting that the stiffer design might be perceived as being tighter.

Under changing loading conditions, $\lambda_{bra}$ also changes. To account for this, the performance metric of pressure (C in Equation (5)) is measured as the average pressure (or $\lambda_{bra}$) of both scenarios:

$$C = \frac{\lambda_{bra1} + \lambda_{bra2}}{2}$$
$$\text{where } \lambda_{bra1} = \lambda_{bra} \text{ for scenario 1,} \qquad (5)$$
$$\lambda_{bra2} = \lambda_{bra} \text{ for scenario 2}$$

### 2.5. Bra Design Generation

In order to create a data set of parametrically defined bra designs to test and evaluate, a generative design method was implemented as a prototype tool in Grasshopper 3D. The tool was designed to facilitate expedient iteration, so that a wide range of variables could be considered. The resulting designs are simple—closely resembling compression bras, they are (mostly) flat, and are expected to have some stretch. They are limited to a front panel, and do not include an underband or back.

The bra generation tool takes in 5 geometries as input: 4 curves, and 1 point, all from a single side of the chest (this tool assumes symmetry); these curves define the base size and fit of the generated bras. In addition to geometry, 6 variables are defined that affect the shape and construction of the bra. These variables (midline position, midline width, strap position, strap width, stiffness ratio, and grid rotation) are diagrammed in Figure 11. The generated bras are reflective of fabric construction by defining a grid, which is orthogonal to itself but can be rotated within the bra (much like fabric can be constructed into a garment at different angles). The variable of "stiffness ratio" describes the difference in stiffness of one direction in the grid to the other, which is similar to the properties that fabric inherits from its warp and weft directions. For fair comparison, the stiffness ratio is applied to each direction as a percentage of a base stiffness which is the same for all generated bras.

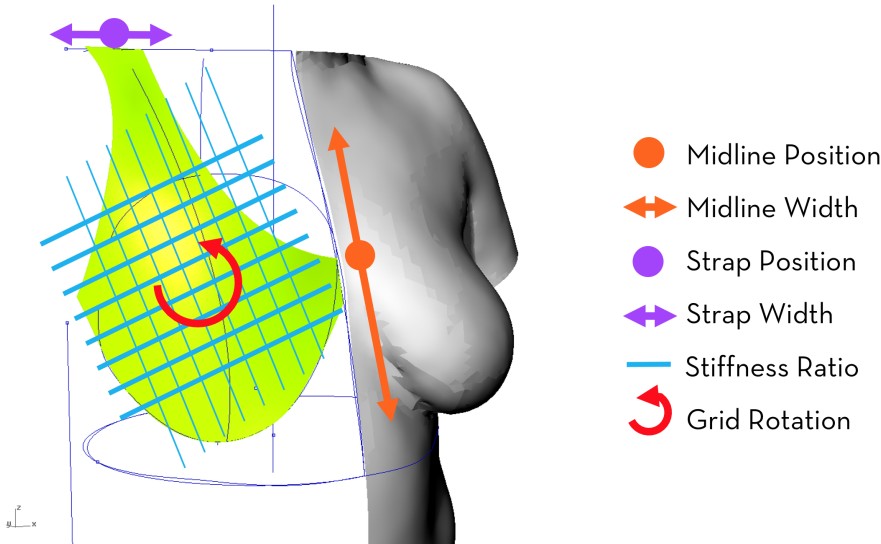

**Figure 11.** Diagram and legend of variables used in bra generation.

Each of the variables described above is set up as a parametric slider in Grasshopper; a numerical component which can output any value of a prescribed decimal precision which is between some preset minimum and maximum. The last step in this generative method is iteration which is implemented using tools from a Grasshopper plug-in called "Design Space Exploration" (DSE) which was created by the Digital Structures group at MIT [26,27].

Using the "Capture" tool from DSE, iteration of bra design generation can be set up and launched over an *n*-length list of design vectors, which sets the value of each slider variable, producing *n* bra

designs. As each bra design is generated, it is also tested using the bra loading method (described in Section ). The numerical data from the design (the inputs and outputs listed in Table 4) is saved into a .csv file. Additionally, an image is saved as a .png which includes renderings of the generated bra design in its neutral (unloaded) state, as well as in the two loading scenarios. A sampling of generated bras in their unloaded state is shown in Figure 12:

**Table 4.** Bra generation inputs and outputs.

| Inputs | Types | Outputs | Types |
|---|---|---|---|
| Underband Curve | Geometry (curve) | Bra Shape | Geometry (points & lines) |
| Underwire Curve | Geometry (curve) | Assigned Force Densities | Number |
| Midline Curve | Geometry (curve) | Rendering | Image (.png) |
| Shoulder Curve | Geometry (curve) | | |
| Armpit Point | Geometry (point) | | |
| Midline Position | Number | | |
| Midline Width | Number (percentage) | | |
| Strap Position | Number | | |
| Strap Width | Number (percentage) | | |
| Stiffness Ratio | Number | | |
| Grid Rotation | Angle (degrees) | | |

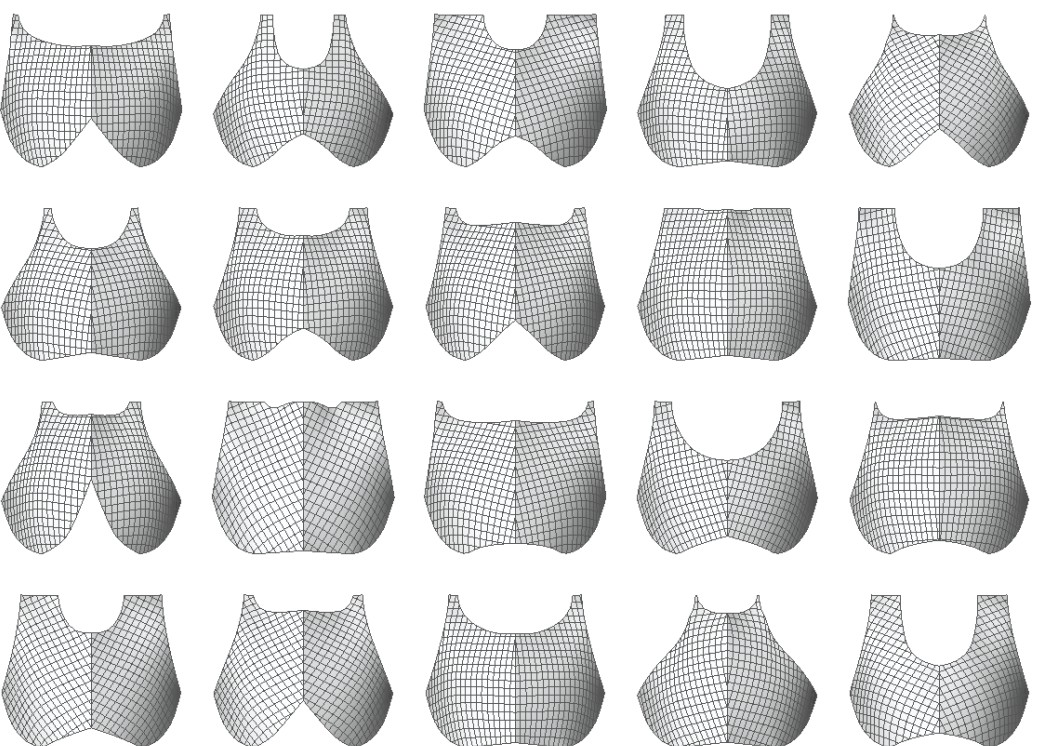

**Figure 12.** Sampling of generated bras.

## 2.6. Variables and Objectives for Evaluation

Each generated bra, once it has been loaded, is represented by the 6 variables that were used to generate it and 2 objectives measured during bra loading, as well as several additional observations that can help us analyze results. These variables and objectives are listed below in Table 5.

**Table 5.** Design space exploration variables and objectives.

| Variables | Objectives |
|---|---|
| Midline Position | Average Pressure (Equation (5)) |
| Midline Width | Average Motion (Equation (4)) |
| Strap Position | |
| Strap Width | **Observations** |
| Stiffness Ratio | Maximum Motion (Between Scenarios) |
| Grid Rotation | Bra Surface Area (Before Loading) |

Periodicity: Angle and Stiffness Ratio

The design variable of angle-describing the rotation of the grid (or fabric) is periodic, in that an angle of 0 (degrees) results in the same design as an angle of 180 (Figure 13). This can make it difficult to interpret patterns in the evaluation results and run principal component analysis. To correct for this periodicity, the angle variable is recalculated to the be sine of the initial angle, which effectively translates to a measure of how close to vertical the angle is (where 1 means the stiffer direction is vertical, and 0 means the stiffer direction is horizontal). While this solves the issue of equivalence at either extreme of the angle variable, it also introduces a new issue of equivalence between the extremes-as **sin**(150) is equal to **sin**(30), even though those angles do not represent equal designs.

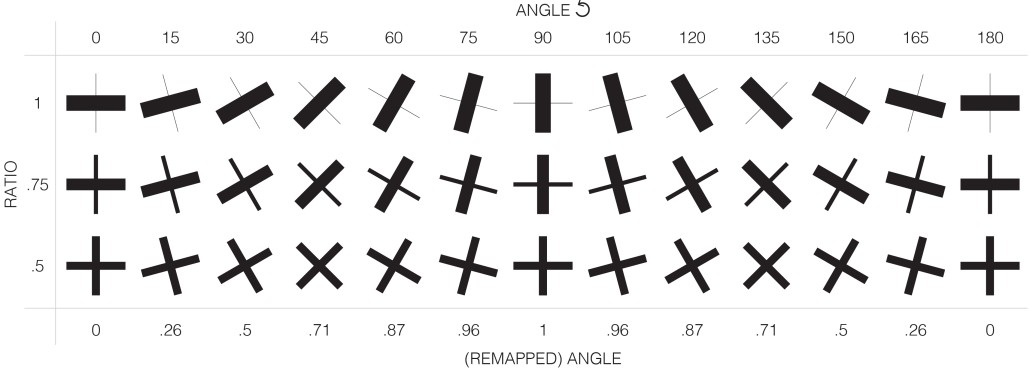

**Figure 13.** Graph showing remapping of ratio and angle variables.

Periodicity is an area of the evaluation process that would need to be addressed in future evaluations of results, especially if exploring loading scenarios at new (non vertical) angles. However, for the vertical loading scenario explored in this thesis, remapping the angle variable and assessing angle as the more simplistic "verticality" still provides compelling insights.

## 3. Results

### 3.1. Results of Body Scan Processing Clustering

The goal of collecting measurements from body scans within this framework is to give designers access to any information they may need in order to create well-fitting sports bras, as fit has been shown to be critical to their perceived comfort and performance. Although this method could be used to create custom designs for individuals, it is uncertain if custom-fit bra manufacturing will be financially feasible in the near future. Discretized sizing—covering a range of body types with a finite number of unique sizes (which each represent a uniquely manufactured product)—is likely to continue to be the norm in bra manufacturing. As a test of the utility of this method towards providing fit criteria for discretized sizing (as opposed to custom sizing), a preliminary machine learning clustering and analysis of the data set of measurements from ~500 scans has been implemented.

For this preliminary trial, the k-means method has been used to cluster over body measurements that most reflect current sizing methods in order to validate this method. Those measurements are

- volume: measured from 3D scan,
- band size: rounded (up to the nearest even number) underband + 4", and
- cup size: rounded overbust-band size

k-means clustering (a supervised-learning algorithm) aims to partition a given *n*-dimensional data set into *k* number of clusters around *k* number of centroids, such that the distance of a data point to the centroid of its assigned cluster is minimized in each dimension. Using squared distances analysis, eight clusters were identified. These eight clusters are represented in Figure 14 in a parallel coordinate plot.

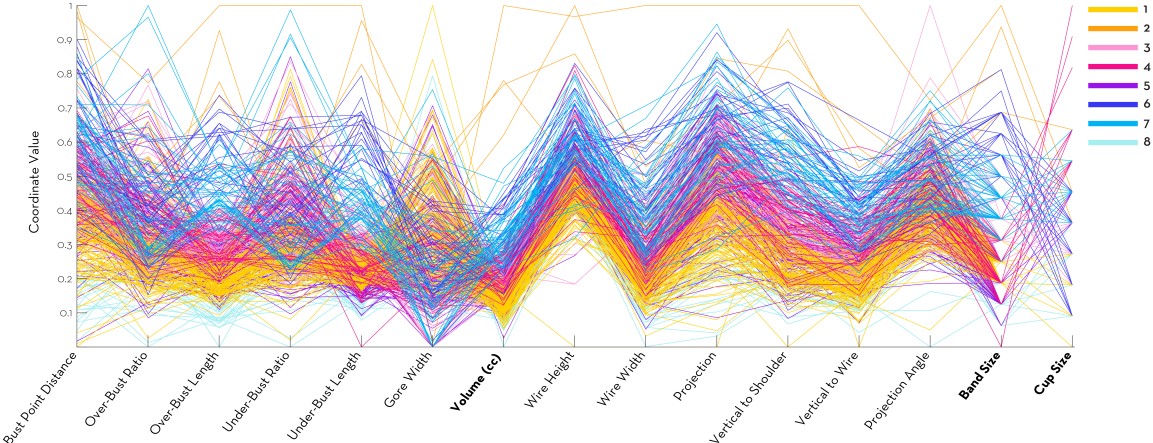

**Figure 14.** Parallel coordinate plot of measurement data, colored according to clusters.

Figure 15 represents the clusters in a different way, creating an average image for each cluster by overlaying the images that belong to it. Looking at these images, it seems that k-means clustering has created groups that can be ordered by increasing size of both breasts and torso. These clusters would provide more nuanced fit than simply "small, medium, large", but it is not clear how they would compare to the more complex band and cup sizing (which can encompass up to 72 sizes) that is typically assigned to everyday bras, or some encapsulated sports bras.

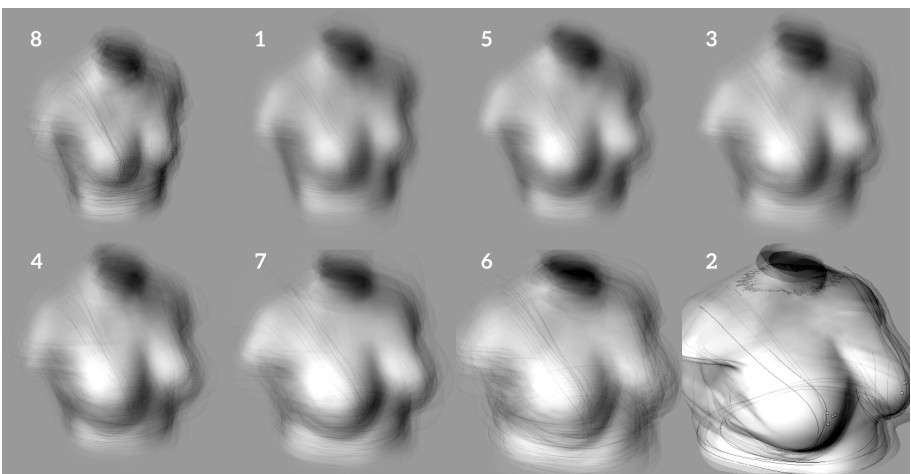

**Figure 15.** Average images for each cluster, arranged by apparent size.

### 3.2. Multiobjective Evaluation of Generated Bras

Of 800 attempted bra designs, 552 designs were successfully generated and tested. Those 552 designs are represented in images (Figure 12) and in a data set containing every design's variable, objective, and observation values. Given all of this data, several graphs were created to try to answer the question: "How do different design properties of sports bras affect their performance?"

### 3.2.1. Patterns in Bivariable Plots

It is likely that some variables have compounding effects on the objectives, but those interactions are difficult to find from plotting single variables at a time. When two variables are plotted against one objective, interactions between design variables and their effects on predicted performance begin to emerge (Figure 16). By fitting a surface to the data points, the significance of that interaction can be perceived from the r-squared fit of the generated surface.

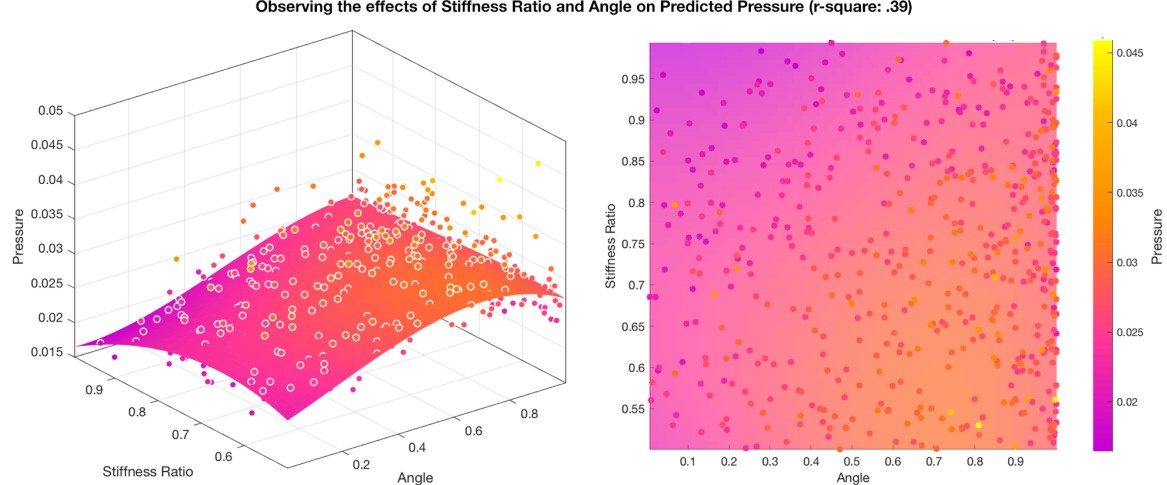

**Figure 16.** Stiffness ratio vs angle vs. pressure with gradient surface overlay.

In Figure 16, this data is shown in three dimensions, as well as in a 2-dimensional projection. This allows for easier, more intuitive understanding of the interactions between the two variables explored in the plot. A brief glimpse at the plot shows that the designs that best minimize pressure are in the top left region (deeper purple), where stiffness ratio is high and angle is low (meaning designs where one direction is notably stiffer, and that stiffer direction is oriented horizontally).

### 3.2.2. Optimal Designs in Biobjective Plot

This data can also be visualized by simply plotting the objectives of pressure and average motion against each other.

Using this plot, the best performing designs (the designs that minimize both performance metrics of pressure and average motion) can be found by searching the pareto front. In Figure 17, the output rendering of each of those best performing designs is included. An interesting trend that is now visually apparent is that as the midline comes up higher at the bottom center of the bra (designs 409 and 168 are such examples), average motion is most reduced but pressure increases. This might be reflective of the impression that encapsulated bras, which are recommended for bounce reduction and support, are also associated with tightness and digging in [11].

By adding color data to the points according to different variables, an assessment can be made if that variable has a strong correlation with performance (of either objective). In Figure 17, the plot is colored according to stiffness ratio. A trend can be seen as the colors in the plot resemble a gradient which suggests that the most optimal designs (the ones along the pareto front) are ones that have a more uneven stiffness ratio (closer to 1).

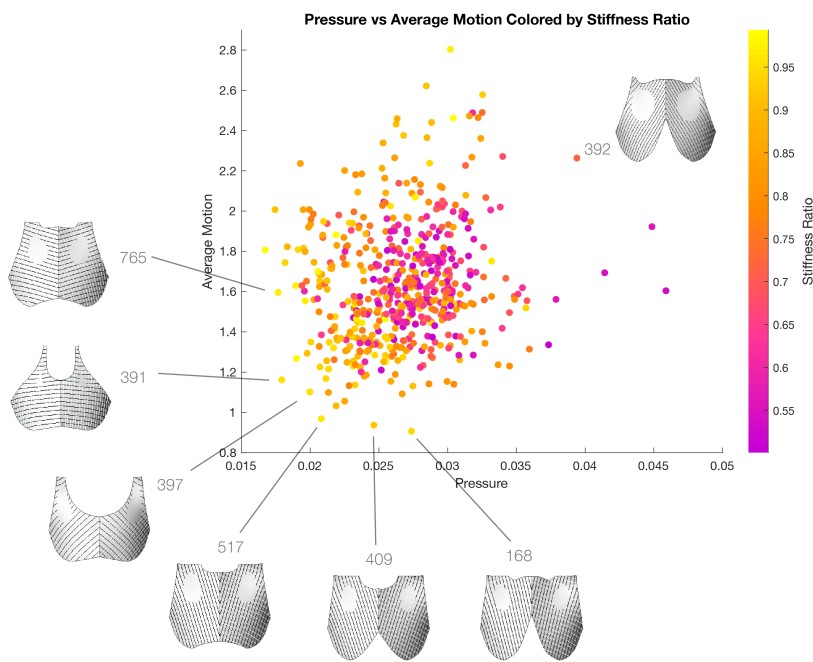

**Figure 17.** Biobjective plot of generated designs.

### 3.2.3. Filtering, Parallel Coordinate Plotting, and Principal Coordinate Analysis

So far, only two or three axes have been considered at a time in these results. Yet another way that this data can be represented is by creating a parallel coordinate plot of all variables and objectives. In Figure 18, the data is filtered to create three groups: the top 10% performing designs in average pressure, average motion, and those that are in top 10% for both objectives.

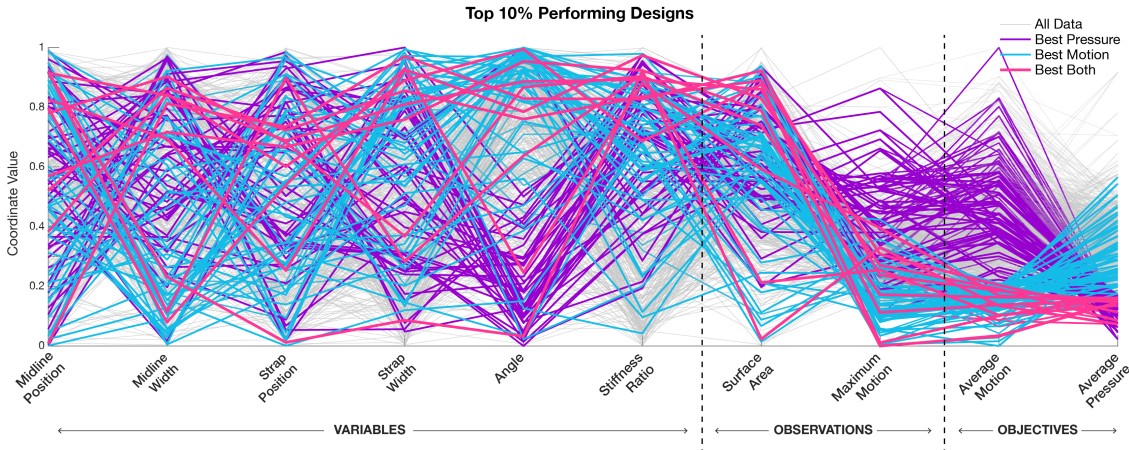

**Figure 18.** Parallel coordinate plot of variables, observations, and objectives with filtered designs.

Investigation of the parallel coordinate plot in Figure 18 reinforces some earlier observations, and provides new ones:

1. Better performing designs tend to have more uneven stiffness (stiffness ratio closer to 1).
2. The designs that perform best in both objectives all have a stiffness ratio of at least 0.7, forming a tighter concentration through that axis in the plot.
3. There is a trade-off between the objectives—most designs that perform better in motion have worse performance in pressure and vice versa.

4. Designs with better pressure performance have their stiffer direction aligned more horizontally (angle closer to 0), versus designs with better motion performance that tend to be more vertical.

5. Angle and stiffness ratio seem to have much more of an effect on performance than the four other design variables (midline position, midline width, strap position, and strap width).

Running Principal Component Analysis (PCA) [28] on just the data filtered for best designs helps identify how these best designs vary the most and least. Figure 19 shows the filtered data remapped to the first two principal components created by PCA. The variance in these two components can account for 57% of the variance in the filtered data. The vectors overlaid on this plot show how the variables have been remapped to this reduced space. This PCA confirms that in these better designs there is high variance in angle, as there are concentrations of best pressure and best motion designs on either extreme. It also confirms that there is less variance in stiffness ratio as designs were mostly concentrated towards the uneven (closer to 1) extreme.

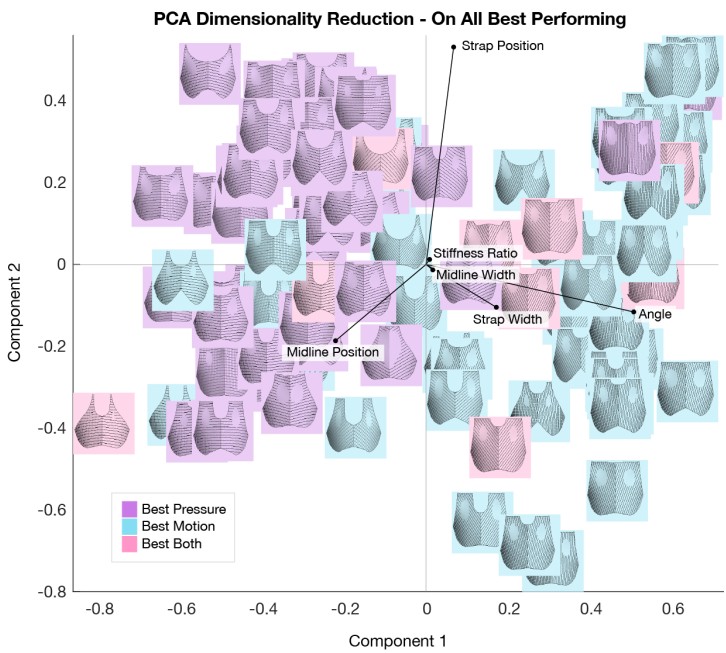

**Figure 19.** Visualization of principal component analysis of filtered data.

## 4. Discussion

### 4.1. Summary of Contributions

In the search to better understand the effect of design parameters on desired performance, this research has created and implemented a framework of informed design methods to explore the sports bra design space with the ability to predict performance. The pieces of this framework are listed below.

- A method for body scan processing, which identifies landmarks on the body and outputs 17 measurements describing size and shape.
- A method for bra design generation, which designs any number of bra designs determined by the design parameters of midline position, midline width, strap position, strap width, angle of rotation, and stiffness ratio.
- A force density method mathematical model, which powers the simulation methods to calculate the response of a mesh to given loads.
- A method for breast simulation, which calculates loads experienced by the breast at given rotations.

- A method for bra loading, which applies the simulated breast loads to every generated bra design, and records their response to represent the objectives of motion and pressure.
- Evaluation methods to visualize and interpret the data collected with the bra loading method.

This paper has shown that this framework can be used to discover insightful guidelines for sports bra designs, as well as propose specific high-performing designs. Use of this framework at the early stage of the design process may help to expand the diversity of designs considered without uncertainty of performance outcomes.

The framework and methods were tested by implementing them through tools to explore the sports bra design space for running. The results of that test suggest that for an exercise like running (primarily creating vertical motion in the breasts), using fabric which has greater stiffness in one direction than the other will increase both support (lowering range of motion of the breasts) and comfort (reducing the overall pressure applied by the bra). The results also show that by orienting the stiffest direction horizontally or vertically, a designer can tune whether the garment will optimize for comfort (horizontal orientation reduces pressure) or support (vertical orientation reduces range of motion).

### 4.2. Limitations and Future Work

#### 4.2.1. Validation

Many of the methods created in this research would be more compelling with validation through experiments. The method for body scan processing could be validated by comparing its output data to that of existing anthropometric software, or to measurements taken in person on subjects. The method for breast simulation has only been validated by comparing its results to a single scan, in one pose. The bra loading method tests bras using a very simple mapping of loads from the breast simulation.

While results of bra loading need not be absolutely accurate for relative comparison, it would be important to show that the tool (and its simple method) accurately predicts relative performance of different designs. This could be done by creating prototypes for several of the generated bra designs and comparing how they rank in pressure and motion on real subjects to how their relative rankings using this framework. However, despite the need for validation throughout the framework, it is encouraging that the two designs that reduced motion the most (seen in Figure 17) also had high midlines and worse pressure performance, which is consistent with observations about encapsulated bras from literature [11].

#### 4.2.2. Increasing Complexity of Bra Designs and Simulation

The bra design generation method creates designs that are representative of very simple compression-style sports bras. Although this simplicity was a necessary first step, it would be possible to add some complexity to the designs generated without having to change the framework. This could include having more complex seams, adding diagonal elements in the grid, or having panels or areas with different stiffness properties. Additionally, the framework does not consider the effect of the underband or the back panel of the bra, and instead treats the seams at the top of the strap and bottom of the front panel to be fixed. While incorporating these elements may be difficult, it would likely provide even more compelling insights for sports bra design.

In addition to the potential benefit of adding complexity to generated designs, there are obvious opportunities for exploring increased complexity in simulation. The bra loading method does not consider friction between the breast and the bra, which may have a significant effect on perceived and real breast motion. This effect would likely be more evident if simulations were time-based to represent dynamic motion. Simplicity was a very conscious choice for the simulation methods in this paper in order to maintain the ability to quickly iterate and compare results; however, it is likely that some complexity could be introduced without overly compromising speed.

### 4.2.3. Simplifying Evaluation

The evaluation methods shown in this paper involve graphing data in many different combinations, resulting in upwards of 70 plots to investigate. While it is an acceptable workflow for research, it may be an obstacle to designers who wish to use this framework in the early stage of their design process. Creating an intuitive interface or new standard terms to describe relationships between variables could remove that obstacle.

**Author Contributions:** Conceptualization, A.B., C.M., and A.E.H.; methodology, A.B.; software, A.B.; formal analysis, A.B.; investigation, A.B.; resources, C.M. and A.E.H.; data curation, A.B.; writing—original draft preparation, A.B.; writing—review and editing, A.B., C.M., and A.E.H.; visualization, A.B.; supervision, A.B., A.E.H.; funding acquisition, A.E.H. All authors have read and agreed to the published version of the manuscript.

**Funding:** This research was funded by Adidas.

**Acknowledgments:** We would like to acknowledge that this work would not be possible without the technical assistance of Yijiang Huang, Renaud Danhaive, and Pierre Cuvilliers.

**Conflicts of Interest:** This research was funded by Adidas, and data for body scan analysis (comprising of 3d scans) were provided by Adidas. However, Adidas had no role in the design of this framework; in the analysis of data or interpretation of results; and they exercised no editorial control over this manuscript, nor did they influence the decision to publish the results.

### Abbreviations

The following abbreviations are used in this manuscript.

| | |
|---|---|
| DSE | Design Space Exploration |
| FDM | Force Density Method |
| FEM | Finite Element Method |

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
