# Peer review of "Body Scan Processing, Generative Design, and Multiobjective Evaluation of Sports Bras"

_applsci, doi:10.3390/app10176126_

Round 1
Reviewer 1 Report
The topic on the development of an engineering design framework to assist the sports bra design is very interesting. The concept is novel and the FDM will speed up the simulation process as compared to the current FEM. Nevertheless, many of the processes are simplified due to the complexity of the framework. It may be quite challenging for academic and industry people to understand. I also wonder if this paper could show model validation with experimental verification. Some key areas must be addressed and clarified.
- Introduction: history of sports bra could be removed.
 - The stiffness of the bar in the membrane can be regarded as the material property of the breast or bra, however, how to determine them?
 - The force densities should not be evenly distributed on the bare breast, it should consider the gravity and the position of mass center.
 - Other than the upward and downward load vectors, does the model consider the interaction between breast and bra in static loading (e.g. gravity)? This interaction may affect a lot in the dynamic movement.
 - The simulation model didn’t validate with experiment results to show the feasibility.

Reviewer 2 Report
Overall this paper is a useful contribution to the knowledge base. It describes a tool to pilot (digitally) the designs of sport bras prior to the creation of physical sample, with the intention of improving the functionality of bras designed. Whilst the research methods are appropriate and the outcome of interest and impactful, the article is not written in way that best justifies or evidences the research.
The article requires: additional detail in the rationale, introduction and methods /results, the language is too colloquial (the use of pronouns is detracting from the communication of the research) and the discussion fails to critique the results in the context of previous studies. By improving the language style used statements will be condensed allowing the required additional detail to be included without too much of an increase in the word count.
Additional detail
Rationale / introduction:
This article lacks scientific detail in its rationale and introduction. This is not to say that the paper isn't warranted, but that the authors could position the article better.
For example, the introduction outlines that women are still experiencing breast pain regardless of developments in bra design. However, this is assuming that bra design is a solution to reduce breast pain - a 'logic' step that links poor bra design and breast pain, to suggest that better design is needed, is missing.
Information about the prevalence of best pain would help readers contextualise the severity of the clinical need that needs addressing.
Another example would be that the article does detail the impact such a tool could have on the wider design process other than time, and on the wellbeing of a woman. Further detail could be added, and previous studies referred to that suggested such a tool is required to streamline the process for environmental reasons and that breast pain is a wide and impactful problem. By not doing so the authors are underselling the value of this research and failing to place the research in context.
Methods /results:
At present the methods are not reproducible due to the absence of essential details and a blurring between what are methods and what are results. Examples of where details should be added include:
Line 123-126 / Table 1: " We used our on a dataset (provided by adidas) of 583 anonymized scans of women aged 18-25. Of these 583 scans, our tool was able to identify landmarks and collect measurements from 472 scans (87%), outputting 17 measurements, as well as an identifying scan number."
- What scanning system did adidas use to collect the scans? Was the scan data raw or had it been post processed? This information is important because it is the quality of the scan data that will determine the quality of the measures (such as volume) that you estimate.
- Of what ethnicity, weight, height etc. were these women? Have you just created and validated a tool based upon a subject of women? This information is essential for a reader to determine how applicable and impact your tool is.
- What are the definitions of the measures you extract in table 1? How one reader would define over bust circumference is completely different to how somebody else would. This ambiguity leads to confusion for the reader and renders the method unreproducible.
Line 191-192: "This patch defines the back wall of the breast".
- How? This isn't an easy thing to do and can be achieved / is defined in many different way.
Line 211-213:"Although this approach may not be exact, it is a reasonable first order approximation if slippage between the bra and the breast is minimal."
-Reasonable according to who or what, and in what context? This language causes uncertainty in what you are saying. This needs to be clearer.
Line 345-346:"The designs that perform best in both objectives all have a stiffness ratio of at least .7, forming a tighter concentration through that axis in the plot".
- Why? Based on what information has this judgment been made? Is it based on previous modelling, biomechanics or self-reported 'fit' studies? The term 'best' is too subjective and clarity need to be provided as to what 'best' means in this context.
The methods and results sections currently read as tool design methods, tool evaluations / evaluation results. Separating the current format into methods; tool design, tool evaluations and the results might be clearer.
Ethics - The acquisition of ethical approval is not mentioned. As the data processed was secondary data it is most likely that ethical approval was not warranted. However, it would be helpful (good practice) to have a statement outlining this and to confirm that this data was acquired ethically and in accordance with the necessary data protection regulations.
Language:
The article continuously uses the first person pronouns such as “our” and “we”. Whilst the use of first person pronouns can be impactful (particularly in presentations) to draw the listeners attention, in this article it could - did for myself - make the result and method section be misconstrued as opinion rather than evidence generated from the methods, and draw attention away from the communication of the research. Thereby introducing uncertainty in the statements that are made. Furthermore, the use of such pronouns is adding unnecessary words which elongates the sentences and further detracts from the statements being made. The authors are encouraged to rewrite the article without the use of first (or third) person pronouns in order to place the focus on the research that is being communication and improve the clarity of that communication.
Discussion:
The discussion needs to not be a serious of statement, but a critical evaluation of the results presented in context of previous literature that includes the critiques the primary projects limitations and outlines areas for future research. This is not yet achieved within this article.
This article has great potential which is not yet achieved.
Specific comments:
Line 123-126 / Table 1: " We used our *WORD MISSING* on a dataset (provided by adidas) of 583 anonymized scans of women aged 18-25. Of these 583 scans, our tool was able to identify landmarks and collect measurements from 472 scans (87%), outputting 17 measurements, as well as an identifying scan number."
Line 242: " Each of the variables described above is set up as a parametric slider in Grasshopper: *SHOULD BE A SEMI COLON* a numerical…"
Round 2
Reviewer 2 Report
Thank you for addressing the comments. I particularly like the introduction.
I would suggest that the authors have another pass through the paper to remove any unnecessary words. For example (although the authors will want to sweep the whole paper):
Line 50 "Although we know that having breasts can impact athletic performance
51 [4,5]" 
Line 52 / 53 "Sports bra research is not yet at a stage that allows us to prioritize athletic performance; instead, we must first tackle the issue of perceived performance must first be addressed"
Line 183 -186 "While a similar analysis may be possible with FDM, our research makes the assumption that instead of examining the complete path of a breast, we can glean similar knowledge is evidenced by from comparing its approximate state at either end of its harmonic motion (at the extremes), producing a predicted "range" of motion."
Line 332 / 333 "Using squared distances analysis, 8 clusters were identified would be a good fit for our data set. "
Author Response
All of the reviewer's suggested changes were made for lines 50, 52, 183, and 332. In following the advice to look through the manuscript once more to remove unnecessary words, we also made minor changes in the following places: Lines 3-6 (the abstract), 60-61, 64-65, title for Figure 1 ("our framework" changed to "proposed framework"), Line 91, 102, Footnote 1 on page 5, Lines 159, 173-174, 179-180, 189, 225, 309, 336, 373-374, 390-391, 406, and 411-412. The authors would like to express sincere gratitude for the thoughtful and rigorous feedback that this reviewer provided. Thanks to this reviewer's comments, the manuscript has improved immensely.